# The Daily Mile in practice: implementation and adaptation of the school running programme in a multiethnic city in the UK

Ash Routen,[1] Maria Gonzalez Aguado,[2,3] Sophie O' Connell,[4]
Deirdre Harrington  [5,6]

[1]NIHR Applied Research Collaboration East Midlands, Leicester, UK
[2]School of Communication, Media and Sociology, University of Leicester, Leicester, UK
[3]Yorkshire Quality and Safety Research Group, Bradford Institute of Health Research, Bradford, UK
[4]Leicester Diabetes Centre, University Hospitals of Leicester NHS Trust, Leicester, UK
[5]Diabetes Research Centre, University of Leicester, Leicester, UK
[6]School of Psychological Sciences and Health, University of Strathclyde, Glasgow, UK

**Correspondence to**
Dr Deirdre Harrington;
deirdre.harrington@strath.ac.uk

## ABSTRACT

**Objectives** The aim of this study was to generate new evidence on how The Daily Mile (TDM), a popular school-based running programme in the UK, is implemented in a diverse and multiethnic city in the UK and also the barriers faced by non-implementer schools.

**Design** Mixed method cross-sectional study (including survey data collection and qualitative interviews).

**Setting** Primary schools in a multiethnic city in the East Midlands, UK.

**Participants** Forty-two schools in Leicester city completed an online survey, and five teaching staff from five schools took part in follow-up semistructured qualitative interviews.

**Results** Overall, 40.5% of schools who completed the survey reported having never implemented TDM, and 96.0% of implementer schools reported delivering TDM on three or more days per week. Reported barriers included space limitations and safety issues, timetabling and curriculum pressures, and pupil and teacher attitudes. Facilitators of implementation were teacher engagement and school culture/ethos, communication of the initiative and substantial delivery adaptations.

**Conclusions** The findings from this study, based on data from schools in a multiethnic city in the UK, suggest that implementation of TDM is variable, and is influenced by a range of factors related to the school context, as well as the characteristics of TDM itself.

## BACKGROUND

The Daily Mile (TDM) is a physical activity programme which challenges primary schools to have their pupils run a circuit of the playground, or similar, for a minimum of 15 min per day during class time. Since instigation in one Scottish primary school in 2012, TDM has 6286 (as of November 2020) registered schools and nurseries in England. A recent data linkage study reported that one in five primary English schools have registered since 2012, and that registration is higher in urban areas, and for schools with higher levels of disadvantaged pupils.[1]

### Strengths and limitations of this study

► Use of mixed methods in a multiethnic city.
► Views from non-implementer schools collected at two stages.
► Sample may represent schools who were particularly motivated to voice their opinion on the implementation of The Daily Mile in their school.
► The focus on one city means that these findings cannot be generalised to all primary schools.

Support for TDM has been such that its use was advocated in schools in the 2016 and 2018 UK Government childhood obesity strategy[2 3] and in 2017 the Scottish Parliament wrote to all schools in Scotland to promote its uptake.[4] Despite policy and funding support, evidence on the TDM effectiveness and context-specific operation is only emerging. A large (2280 children from 40 primary schools) randomised controlled trial conducted in Birmingham, England, examined effectiveness of TDM and found no significant difference (p=0.146) in the change in the primary outcome (BMI z-score) between intervention and control schools after 12 months. The accompanying economic analysis did report TDM to be cost effective.[5]

Regional qualitative and ethnographic studies have examined how schools implement TDM and what facilitates or constrains delivery from the perspectives of head teachers, teaching staff, pupils, parents and school governors as well as stakeholders including public health practitioners. These were conducted in Scotland,[6 7] Wales,[8] South London[9] and rural England[10] and represent the views of schools from a variety of deprivation and geographic locations.

These studies have identified a range of factors associated with implementation of TDM including simplicity of the intervention,

flexibility of delivery, time constraints, weather and the school physical environment, impact on learning time, teacher participation, whole-school delivery, use of goal setting, adaptation and embedding into regular practice. Although a wide variety of themes have been identified, the qualitative investigations of delivery and implementation[6–10] have largely focused on perceptions of teachers and schools who currently deliver TDM. There is a gap in knowledge from schools who have chosen not to implement TDM, or those who have tried with limited success.

The aim of this evaluation was to generate new evidence on how TDM is implemented in an ethnically and socioeconomically diverse city and also the barriers faced by non-implementer schools. The objectives were to describe (1) TDM adoption and reach; (2) to uncover key barriers and facilitators to TDM implementation, and (3) to identify other key issues relating to how TDM was being implemented in Leicester, UK.

## METHODS
### Study design
We undertook a mixed-method two-stage evaluation during the 2018/2019 and 2019/2020 academic years. Stage 1 consisted of a city-wide cross-sectional online survey, while stage 2 was an interview with primary schools who completed the first survey.

### Setting and population
Leicester (a multiethnic city in the Midlands, UK) is a microcosm of the UK due to its socioeconomic diversity and proximity to suburban and rural areas. TDM in primary schools is supported by Leicester City Council strategies and schools are encouraged to sign-up via the City Council intranet and through local primary teacher meetings.

### Patient and public involvement
This work was informed by the Public Health team at Leicester City Council, and recruitment was supported by the Leicester City School Sport and Physical Activity Network (SSPAN). No other patient or public involvement took place.

### Stage 1: city-wide survey
The TDM coordinators (or Physical Education leads) of all city primary schools were emailed by SSPAN with the initial online survey link, and the survey was also advertised on local authority schools intranet and at teacher meetings. There were eight questions on TDM implementation and experiences, if any, of TDM. For example, respondents were asked on how many days in a typical week is TDM run in their school and what year groups it is delivered to. Schools reporting no delivery of TDM were asked for reasons for non-delivery. Respondents could add free text comments, and they were asked whether they would be happy to be contacted for a follow-up interview.

Schools that had adopted and utilised TDM (ie, implementer schools), and schools that initiated TDM but had failed to maintain implementation (ie, non-implementers) were invited to Stage 2.

### Stage 2: follow-up interview
The contact person in 'implementer' schools were invited via email to a face-to-face interview. In order to maximise participation of those from Stage 1 who agreed to be contacted for Stage 2 the research team phone called the school reception, sent multiple emails and visited the schools within walking distance of the university in person. Interviews were conducted in schools by one trained researcher (AR) following a semi-structured topic guide. Interviews were recorded on a digital recorder and transcribed verbatim by an external transcription company. Transcripts were anonymised and identifiable information removed.

### Survey and interview guide
Details of the survey and interview questions are included as online supplemental files 1 and 2. The survey was designed pragmatically to gather information on the adoption of the programme, the level of implementation and any barriers to implementation. The intention was to use this information to inform TDM support offered to city schools across by SSPAN. As such the survey was not informed by a theoretical framework. The interview questions were designed to query topics highlighted through discussion with SSPAN and informal conversations with local teaching staff. Quesstions related to key themes of the RE-AIM framework such as Reach and Maintenance,[11] and the programme components themselves (eg, delivery of Daily Mile core principles) were also included.

### Analysis
Survey data are summarised as percentages and ranges. No inferential statistical analyses were conducted on these data. A thematic analysis[12] was completed by one trained qualitative researcher (MGA); this was chosen to explore interviewees' perceptions on the implementation of the TDM at their schools. An inductive approach to analysis was followed. Transcripts were read and re-read to identify relevant features. DH supported qualitative data analysis by participating in the definition and refinement of themes to enhance data reliability. Interview data were systematically coded at the descriptive and semantic level using Nvivo (V.11.4.3).

### School data
School sociodemographic characteristics (size, geographical location, ethnic mix) were taken from School Census records (January 2019). Deprivation data of the ward (locality division for electoral and planning purposes) in which the school is situated is based on Index of Multiple Deprivation (IMD) 2019.

**Table 1** Characteristics of the schools involved in Stage 1 (n=42)

|  | Implementers (n=25) | Non-implementers/ other (n=17) |
|---|---|---|
| Pupil number (range) | 401 (178–608) | 440 (108–747) |
| % BME pupils (range) | 49.8 (21.5–97.8) | 71.5 (20.2–98.0) |
| % FSM (range) | 23.4 (5.1–44.9) | 17.9 (108–758) |
| % in IMD Q1 (range) | 40.0 (0–100) | 24.5 (0–61.1) |
| % in IDACI Q1 (range) | 48.9 (10.8–100) | 35.7 (9.2–69.6) |

BME, black and minority ethnicity; FMS, free school meals; IDACI Q1, Income Deprivation Affecting Children Index Quartile one (most deprived); IMD Q1, quartile one (most deprived) of Index of Multiple Deprivation.

## RESULTS

At the time of the research (winter 2019), there were 83 primary schools available to participate in TDM in Leicester city. Of these, 42 (50.6%) completed the stage 1 survey (table 1). Of these, 19 schools responded with willingness to take part in the interview (stage 2).

### Implementation level and type

Of the schools that completed the survey, 40.5% (17/42 schools) reported never running TDM. Of those who did implement it (ie, 59.5%; 25/42), 56.0% (14/25) reported running it 5 days/week, 4.0% (1/25) on 4 days, 36.0% (9/25) on 3 days, and 4.0% (1/25) on 2 days/week.

Table 2 presents the proportions of schools delivering TDM to various combinations of year groups. For example, 69% (20/29) of schools implementing TDM deliver it to all year groups from foundation year, to year 6. Schools reported in the survey that TDM delivery 'varies for different year groups' and the 'amount actually varies greatly between classes and teachers.'

The survey free text responses provided relevant insight on the schools' reasons for not delivering TDM. Four participants reported that their schools were actually implementing an adaptation of TDM: 'We do a less formal version where classes go for a walk, jog about 3 times per week'. On the question 'Why does your school not deliver TDM?' eight participants reported difficulty in delivery and/or resource or staffing problems. One school mentioned parents' lack of support to the initiative as a reason not to deliver TDM.

### Stage 2: qualitative themes

Interviews were conducted with TDM coordinators at five primary schools (see school characteristics in table 3). Two of the five schools either did not deliver TDM or had previously delivered and discontinued after a few months. One of the schools delivered TDM before school at a local cricket ground (which they pay for) and parents sometimes watch the children.

Themes emerged through the aggregation of representative subthemes that reflected the research questions, as well as new emergent features in the data. The iterative comparison of the responses in the interviews resulted in a coding framework consisting of five themes: 'Barriers to the delivery of TDM', 'facilitators of the TDM', 'tailoring physical activity offerings', 'perceived benefits' and 'measuring TDM success.' The description of these themes and the content of subthemes are below.

### Barriers to the delivery of TDM

#### Characteristics and functioning of the school

Three types of barriers related to the characteristics and functioning of the school were identified: space and safety concerns, pressures to deliver the National Curriculum and timetabling difficulties.

Four of the five interviewees affirmed that the lack of a safe and appropriately sized playground or field complicates the running of TDM, while three reported not having adequate space for the optimal running of TDM. Safety concerns relating to the quality of the space where pupils run also emerged as a possible implementation barrier. These concerns were related to the safety of the playground during adverse weather (for instance, puddles or icy pavement), that some pupils from disadvantaged families do not have appropriate footwear for all weather conditions and the risk of having a high number of pupils running at the same time in a small playground:

> (…) obviously, if we took a few classes down at a time health and safety becomes a little bit of an issue because we've got benches in the way, we've got bits around the playground so trying to get the boys not to be racing round and the girls.
>
> (School 1)

The description of space as an important barrier resonates with the free text qualitative data from non-implementer schools in stage 1. These schools reported lack of space as one of the primary reasons for not delivering TDM. Some schools, especially those with larger pupil numbers, feel that they do not have enough space to deliver TDM: 'Due to the space and the size of the school.

**Table 2** Proportions of schools delivering TDM to year groups

| Year groups | Foundation | Year 1 | Year 2 | Year 3 | Year 4 | Year 5 | Year 6 |
|---|---|---|---|---|---|---|---|
| % | 71.9 | 75.0 | 81.3 | 87.5 | 90.6 | 87.5 | 87.5 |

TDM, The Daily Mile.

**Table 3** Characteristics of the schools involved in stage 2

| School | School pupil number | % BAME | % FSM | Deprivation of city ward | TDM delivery | Contact person's role within school |
|---|---|---|---|---|---|---|
| 1 | 596 | 51.5 | 7.4 | Quintile 4 | No, 'it was done sporadically for a few months' | Higher level teaching assistant and PE coordinator |
| 2 | 379 | 59.4 | 25.3 | Quintile 1 | Yes | Senior management team |
| 3 | 366 | 24.6 | 25.1 | Quintile 1 | Yes | PE coordinator and class teacher |
| 4 | 235 | 49.8 | 20.4 | Quintile 2 | No, The Daily Boost as an alternative is on-going | PE and history coordinator and class teacher |
| 5 | 468 | 40.6 | 23.4 | Quintile 2 | Yes, although The Daily Boost is used with years 1 and 2 | PE Teacher |

Deprivation of city ward is based on IMD where a lower quintile is the most deprived.
BAME, Black, Asian and Minority Ethnic group; FSM, free school meals.

We only have a playground and 480 children. When PE is on there is a lack of space due to safety.'

A second barrier within this theme is the teachers/ pupil timetables. Four schools found it difficult to fit TDM in their timetables without negatively impacting other teaching duties, such as assessments and lessons. In addition, delivering TDM on a daily basis makes one of the lessons fifteen minutes shorter.

> Well, in a week where you just think ahh, I've got so much to get through and we spent longer doing this so now we're running behind and it feels like that a lot of the time for a lot of us that we're always trying to fit more in than really we can. So, for some people. The Daily Mile is a thing too far, sadly.
>
> (School 2)

Stage 1 participants also highlighted the difficulties including TDM in the school's timetable as a barrier to delivery: 'Teachers timetables are so full with regular lessons and other initiatives that The Daily Mile would be too much of a burden to ask them to fit as well' and 'They (pupils) already do a lot, don't know where it would go in the timetable.'

The third barrier within this theme was the pressure to deliver the National Curriculum. TDM coordinators intimated that delivering the curriculum is more important than pupils' daily physical activity, which could constrain TDM implementation. But it is important to note that two interviewees contend that TDM's impact on curriculum delivery is minimal, as having a physical activity break helps pupils focus. The excerpt below expresses interviewees' perception that the pressure to deliver the curriculum might be a barrier to the adoption and implementation of TDM:

> Even four-year olds are being baselined now [inaudible] timetables and all of this is like ways in which school will be held accountable by Ofsted [inaudible]. So, things like The Daily Mile go on the back-burner, don't they? Which is a shame.
>
> (School 4)

### Pupils' and teachers' attitudes

The second type of barrier related to pupils' and teachers' attitudes towards physical activity, the outdoors and TDM. Interviewees considered that teachers' personal attitudes towards physical activity and being outdoors influences the implementation of TDM:

> I think that is the biggest barrier [school's space] and then some people's lack of enthusiasm for being outside and being active which you get quite…if you have a range of staff that's what you're going to get
>
> (School 2)

### TDM in practice

The main subtheme was that TDM is 'not as simple' in practice. Although interviewees agreed that the TDM is easy to implement, three affirmed that it is not exempt of costs and implementation difficulties. They suggested TDM takes more than fifteen minutes, and involves additional costs for TDM to run autonomously. As outlined in TDM core principles,[13] the TDM has to be quick to deliver. However, the estimated 15 min delivery time does not include time needed to shepherd pupils to the playground, especially if the classroom lacks direct access to the open area. One teacher suggested the real implementation time can be up to thirty minutes.

Running TDM may involve additional costs. Here, interviewees were not referring to small expenses, such as

footprints (painted on the playground to guide children) or tokens to count laps, but to staffing costs and teachers' time after work for additional preparation that optimal, engaging and long-term implementation requires. Interviewees explained that they invested time outside their working hours to collate pupils' times or create pupil progress certificates. One of the interviewees reported that TDM delivery required the collaboration and practical support of numerous teachers and other school staff:

> It does need managing so we have members of staff out. So, dinner ladies, lunch time supervisors and the class teachers now come out to manage because people can get into little groups and cause little bits of aggro so we try and nip that in the bud as much as possible.

> (School 5)

### Facilitators of TDM delivery
#### Schools' existing health and sport culture/ethos and teacher motivation
The approach of the school to health and physical activity can facilitate the long-term success of TDM in a school. All interviewees agreed that the TDM builds into, and contributes to, the wider culture or ethos of health and well-being within the school. Four interviewees described TDM as part of the school's health and sports culture, as an element of the 'school's values':

> Yeah, part of our values is that actually being active and healthy is important and I think we do promote being healthy, our children get an opportunity to do cooking which they don't in a lot of schools, which is great, and we promote healthy eating through that but I think just being active, encouraging that would be, yeah, for it [TDM] to be part of our school values would be good if it was a greater part of our school values.

> (School 2)

Interviewees mentioned several times the importance of the support from the school in running TDM. By support from the school, interviewees referred to the schools' interest in introducing TDM and the acknowledgement that pupils' daily physical activity and the TDM are an important part of the school's values and activities. Teachers' motivation and the development of a health and sports culture within the school can positively influence TDM delivery. Interviews responses showed that teachers' involvement in the delivery of TDM impacts pupils' performance. Teachers running alongside and cheering pupils was reported to be a motivator and aid for pupil engagement.

#### Communication of TDM
Interviews revealed that communicating the TDM is a key to facilitating implementation. Schools make use of their communication channels (eg, newsletters), to communicate that they are running TDM. For some schools, the school's weekly assembly is a key communication space to engage students with TDM and reinforce the role of TDM as part of the school's health culture. During the school assembly, for instance, weekly results of TDM competition (see 'Keep it fresh' below) are communicated and progress certificates handed out:

> (…) if they're doing the right thing then they'll keep going and some children do eight laps, some children do two and at the end of the week I record it all, put it on a spreadsheet so I've got the data of who is doing what, when. We look at celebrating that in assembly with the class that goes the furthest.

> (School 5)

#### Ability to modify limited space
Interviewees explained that small modifications in playgrounds to mark the path or giving pupils a token to count laps can facilitate TDM. One interviewee reported that her school have painted footprints on the playground to make running TDM more intuitive. Interviewees regard the use of tokens and clickers to count laps as positive:

> Yes, something I hadn't thought of, but think could be quite useful, sometimes if children could have clickers, you know when they count people into things, so if they could click every time, they do a lap that might be good in terms of resources.

> (School 2)

#### 'Keep it fresh'
As noted above, if schools deliver TDM according to the core principle of simplicity (getting pupils to run or walk for fifteen minutes per day) pupils 'would lose interest in it after a week quite quickly.' All the interviewees explained that they had to introduce some variations or adapt TDM to maintain pupils' engagement with the initiative in the long-term. Coordinators developed diverse strategies to maintain pupils' motivation and interest in TDM. Strategies included introducing an element of competition (interclass competitions), monitoring and rewarding pupils' progress and *keeping TDM fresh*. As one interviewee explains, they made TDM complicated to engage pupils:

> I've made it complicated to suit what I want to get out of it. If I want my children to do something I have to have that element of competition (…) So yeah, I'd agree with that and I just make a rod for my own back to make it more enjoyable for the children, I guess. For my children because a sleepy village in Scotland has a different cohort to inner city Leicester.

> (School 5)

Monitoring pupils' times and distance ran was regarded as motivating for pupils by coordinators. It is necessary to clarify that that monitoring does not aim at promoting individual competition or creating an 'elite' group of

fitter pupils, but to encouraging self-improvement by presenting TDM as a collective endeavour. Regarding competitivity among pupils, it is important to note that the school 1 interviewee contended that competition does not always work, as many pupils are not competitive.

To ensure pupils' interest and engagement with TDM in the long-term, interviewee number 5 explained that teachers and schools have to 'keep TDM fresh' to motivate and engage pupils. 'Keep TDM fresh', in this case, means updating the initiative on an annual basis, so pupils maintain their interest and motivation year after year. 'Keeping TDM fresh' requires modifying the initiative at two levels: (1) Introducing changes in the TDM implementation design for the academic year (ie, measuring the distance run and translating that into the distance between two cities on a map; introducing competition between smaller groups within a year group and (2) measures to implement TDM on a daily basis (ie, monitoring pupils time, use of tokens, mark the distance pupils have to run).

### Tailoring the physical activity offering

Schools are interested in delivering physical activity programmes to keep their pupils active daily. Schools adapted programmes to overcome internal barriers and engage pupils. Four of the five interviewed schools combine different daily exercise programmes overcoming spatial, timetable and curriculum barriers.

Two interviewees introduced two alternative physical activities to deliver instead of TDM in adverse weather in the absence of a covered playground. Another school used different physical activity initiatives in different school years according to pupils' age to create a culture of health and sports in the school. In this school, younger pupils did The Daily Boost—an initiative encouraging 15 min of organised physical activity at the school, this can be running, walking, cycling, dancing or any physical activity pupils may like—while TDM was implemented among the older pupils:

Oh yeah, so we rolled out The Daily Boost to kind of…so The Daily Mile was running quite happily and then we've gone well, we need to do something for the younger children to get them. So, it's not just a shock at Year 3, so it slowly integrates into their school life. So yeah, we have started to roll that out and that was rolled out at the start of this year.

(Interview, 5)

Schools create their *'own version'* of TDM to successfully implement the TDM according to the necessities and characteristics of their schools and pupils:

Yes, so we've created our own version to keep the children engaged. You see with the skipping, there's so many different variations. They can learn tricks; they can do spins. Like we've got a few big ropes so they're doing different games, different activities with the big ropes, they just love having little competitions against each other to see who can skip for the longest, it's just

a little bit more, they can vary it so much whereas with the Daily Mile you can't really, there's not that much to vary apart from saying to them let's see if you can increase your time but again.

(School 1)

The interview findings reflect the Stage 1 survey where schools, despite the mentioned barriers, have tried to develop strategies to help their situation. For instance, one school applied for funding to build a track to improve their space and another school used skipping ropes to ensure their pupils could still be active in their small playground space in a similar fifteen-minute window. Schools implementing The Daily Boost felt it offered flexibility for their size and space.

### Perceived benefits of TDM

All interviewees reported that TDM benefits their pupils. The benefits can be classified into physical and behavioural. For physical, it is through the improvement in fitness or by encouraging them to practice sports outside the school. Two teachers reported that TDM improved the physical condition of their pupils and increased their physical activity levels.

Interviewees 1 and 5 explained that TDM acted as a catalyser of pupils like for being physically active and sports and making healthier choices. Interviewee 5 tells that one pupil started to become interested in practising sports after school as a consequence of the implementation of TDM at the school:

So, I've got one family who she started doing cross country in Year 3, she's won the medal, so now she goes running with her mum and I didn't expect that to happen and now she loves it, she loves running and she's been successful at it and it's kind of given her an into other sports. Now she's part of the girls' football team, she plays cricket, she gets stuck in with everything, and the running was the catalyst.

(School 5)

The interviewees all said that TDM not only have physical benefits for pupils, but it also brings about behavioural and social benefits. Four interviewees agreed that running TDM improved pupils' attention and concentration. As they explained, having a break for physical exercise helped pupils refocus on their learning. Interviewee 1 also reported that TDM help pupils to manage their anger:

(…) we have a few children that perhaps have behavioural problems, anger problems and you know, they go out and it does help them. They'll say to you that it helps them, and it helps them focus a little bit more if they have that break, go out, do that, come back in and they feel that…

(School 1)

## Measuring success

Although interviewees reported the TDM to be beneficial, there is no evidence of the benefits of the TDM beyond teachers' own perceptions. The interviewees' focused on 'keeping pupils active' putting the ways in which pupils do TDM in a secondary place. The interviewees described TDM as beneficial without giving excessive importance to the fact that some pupils (especially girls) walk instead of running or if they run in smaller laps if the playground is too busy. Interviewees described TDM as a successful initiative because it helps them to ensure that pupils are physically active ('The whole thing is keeping them active'):

> The most important thing for me is that it's done. If we're really squeezed for time and to be honest, sometimes we are and I'll say right, we're going to do half a mile.
>
> (School 2)

## DISCUSSION
### Main findings

Over half of schools in our survey have adopted TDM in some form while 40.5% who completed the survey reported never implementing TDM. In relation to dose delivered, 56.0% of schools who adopted TDM ran it every day which was a TDM recommendation at time of data collection. Subsequently, TDM changed this recommendation to delivery on at least 3 days, which nearly all schools (96.6%) met. The majority (69%) report delivering TDM to all year groups from foundation year to year 6, suggesting there is good reach across the school. However, teachers acknowledged delivery varies within and across year groups.

Our results from an ethnically and socioeconomically diverse city make a useful addition to the extant literature. Although it was highlighted that what works in a rural area in Scotland may not work for a multiethnic city like Leicester no themes related to ethnicity emerged in our study. However, stage 1 data show more of the implementer schools were in the lowest deprivation quartile than non-implementer schools which supplements data showing that TDM-registered schools had more disadvantaged pupils.[1]

TDM coordinators conveyed a range of class, teacher and organisational level issues they felt constrained the delivery of TDM including space limitations and safety concerns, national curriculum pressures, and timetabling difficulties. These themes shed light on how TDM core principles[13] of TDM being quick and simple may not always be possible. In relation to pupils and teachers, their attitudes toward physical activity and the outdoors appeared to act as a constraint to implementation. Facilitators of delivery included coordinating teachers' motivation, and the existing school ethos and culture regarding physical activity and well-being. Finally, the need to adapt the delivery of TDM to maintain pupil motivation was identified as an important issue, as was the creation of alternative physical activities to supplement TDM, or to deliver in place of TDM, due to the needs of the school context.

### Strengths and limitations of the study

Strengths of this study include the use of mixed methods, with the survey providing information on the implementation of TDM across a range of schools in a multiethnic city. These findings have been used to develop a 'top tips' infographic for schools or coordinators working to support schools in collaboration with those supporting TDM at Leicester City Council (online supplemental file 3). Only one extant study has examined TDM implementation at a city/borough level.[9] This enabled identification of variation in practice, including schools who did not implement TDM, which was further examined by following-up a small number of schools for more detailed understanding via interviews. A limitation of this approach is that our interview sample may represent schools who were particularly motivated (schools self-selected availability for inclusion in further research in the survey) to voice their opinion on implementation of TDM in their school, that is, successful implementer schools, or conversely those with strongly held views on the difficulties or non-implementation. A limitation is the response rate of 50% for the online survey which could introduce non-response bias. Difficulties in accessing school staff is not uncommon[14] and practical aspects, such as limited availability of respondents, are explanations for limited sample sizes.[15] The interviews provided rich and nuanced codes and 'meaning saturation'[16] was achieved. The obtained data met additional quality criteria in qualitative research[17]: codes were supported by a wide range of evidence from data and findings resonated with the literature. Therefore, we do not believe the number of interviews affects the validity or generalisability of our findings.[15] However, as is inherent within qualitative work, and the focus on a densely populated urban area, our findings cannot be generalised to all primary schools.

### Comparison to existing literature

The adoption rate of 59.5% is slightly higher than the 48% reported from a TDM evaluation in London schools.[9] However, our value is based on the survey respondents only so it is likely this number is lower. Two previous studies have detailed the dose of TDM delivered in schools. Focused on one primary school in the East Midlands, Harris et al[10] reported key stage 2 teachers delivered TDM on 86% of days, with 95% of pupils receiving the intervention on those days. Hanckel et al[9] qualitatively summarised dose information, suggesting TDM was not delivered every day in most schools/classes, and that it was viewed as interchangeable with other physical health interventions within the school. At the time of writing to our knowledge, there were no other existing data on reach of TDM across year groups.

The barriers of space limitations, timetabling and pressures to deliver the curriculum are consistent with findings from previous TDM evaluations[7–9 18] as well as being common barriers to the delivery of other physical activity initiatives or policies in schools.[19 20] Likewise teacher buy-in (ie, whether they thought it worthwhile and whether they liked it or not) has been reported as a barrier in other school-based running programmes.[21] In relation to implementation facilitators, previous work on TDM has identified that a teacher's active participation in TDM is important for enabling pupil engagement with the initiative[8] The importance of TDM aligning with the existing school culture and ethos towards physical activity and well-being, which acts as a facilitator to implementation, has been highlighted in research on a similar school-based running programme in the UK.[21] Schools' substantial adaptations of the TDM and the development of rebranded, alternative physical activity offerings resonate with a recent qualitative study that highlighted the malleability of TDM.[18] Adapting TDM to counteract a drop off in activity levels[22] and to keep pupils engaged and reduce boredom has happened through offering a variety or choice of activities.

## Implications for practice

It is clear that adaptations and alternatives to TDM are happening. This has implications for TDM evidence base as well as practice. Schools may begin using TDM but adapt or change their approach, often quite substantially, to suit their particular context. This has implications for the relevance of any evidence on the effectiveness of TDM in a real-world setting, as some schools may not necessarily be delivering TDM even if using TDM nomenclature. When promoting TDM to diverse schools within a local authority area, strategies to support schools appropriately adapt TDM or even create alternatives to keep pupils active for fifteen minutes according to the needs and preferences of their school are warranted. The schools involved herein did not mention a need to adapt based on ethnic make-up of the school. One unique point related to disadvantage emerged whereby pupils' shoes were not suitable for all weathers. While a TDM core principle is that children should not need specialist kit and can do it in their school clothes it is conceivable that children in light or low-quality shoes may find doing their mile on wet tarmac uncomfortable. Equally, TDM risk assessments state that children in unsuitable footwear should walk[23] which also has implications for the pupil's own experience and when evaluating the TMD at the pupil level as there are different health and social effects to running and walking a mile. It is important that schools and stakeholders consider factors that support long-term implementation prior to starting TDM in a school. In particular consideration should be given to the existing implementation context within the school, such as the physical environment (space, playground options) organisational issues (timetabling, curriculum focus/pressure) staffing (teacher attitudes and motivation), as well as the

schools culture and ethos regarding physical activity and well-being. These considerations are also relevant to any physical activity programme that is embedded in schools. It may also be important to emphasise to schools that in spite of apparent simplicity, TDM does require teacher time and additional effort to maintain pupil interest and engagement, and it may involve a collective effort involving staff from across the school until it can run autonomously. Finally, as is the case with Leicester city schools, it is clear there are ways to adapt TDM to facilitate delivery and allow children to participate in a vital fifteen minutes of physical activity that it distinct from PE and is embedded in curriculum time.

## Conclusions and future research

The findings from this study, based on data from schools in a multiethnic city in the UK, suggest that implementation of TDM is variable, and is influenced by a range of factors related to the school context, as well as the characteristics of TDM itself. Future research should focus on understanding how to support the readiness of schools for implementation of TDM or other school-based running programmes. Learning from schools that implement adapted versions of the TDM, or alternative programmes with similar core principles would also be valuable.

**Acknowledgements** We thank Daxa Ralhan from Leicester City Council and Sarah Lansdowne from Leicester City School Sport and Physical Activity Network for assisting with school recruitment. We thank Andrew Fry and Anna Harding from the University of Leicester for administering the funding.

**Contributors** AR, DH and SOC conceived the study design and methods. AR collected the data and MGA led the qualitative analysis. MGA and DH discussed and refined the emerging codes and final themes. AR, DH and MGA wrote the first draft of the manuscript. All authors reviewed the manuscript and provided feedback and comments. All authors read and approved the final manuscript.

**Funding** This work was supported by a grant via the Wellcome Trust Institutional Strategic Support Fund at the University of Leicester and funding under the Cities Changing Diabetes programme in Leicester. Cities Changing Diabetes Leicester is a joint working project that is funded by and developed in collaboration by and between Leicester Diabetes Centre and Novo Nordisk. This work has been run by researchers from the University of Leicester, Leicester Diabetes Centre and the NIHR Applied Research Collaboration East Midlands.

**Competing interests** None declared.

**Patient consent for publication** Not required.

**Ethics approval** Ethics approval was obtained from the University of Leicester College of Life Sciences Concerning Human Subjects (non-NHS) application number 19792-dh204-ls:healthsciences. A consent statement was included on the opening page of the online surveys. Individuals being interviewed provided written informed consent.

**Provenance and peer review** Not commissioned; externally peer reviewed.

**Data availability statement** Data sharing is not possible as a statement to allow data sharing was not included within the consent statements. However, all reasonable requests for a collaborative reanalysis will be considered. Please contact the corresponding author for more details.

terminology, drug names and drug dosages), and is not responsible for any error and/or omissions arising from translation and adaptation or otherwise.

**ORCID iD**
Deirdre Harrington http://orcid.org/0000-0003-0278-6812

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
