## [Reviewer comments · BMJ Open]

ARTICLE DETAILS

TITLE (PROVISIONAL)	The Daily Mile in practice: Implementation and adaptation of the school running programme in a multi-ethnic city in the UK
AUTHORS	Routen, Ash; Gonzalez Aguado, Maria; O' Connell, Sophie; Harrington, Deirdre

VERSION 1 – REVIEW

REVIEWER	Pallan, Miranda University of Birmingham, Public Health, Epidemiology and Biostatistics
REVIEW RETURNED	13-Dec-2020

GENERAL COMMENTS	Thank you for allowing me the opportunity to review this paper. I read it with interest as the Daily Mile intervention has attracted a lot of interest at the national level. I have a number of points and concerns about the paper, and I have outlined these below. A major concern is the small scale of the study, which creates very substantial limitations. 1. Abstract – the first sentence does not make sense: “Overall, 37% of schools who completed the survey reported having never implemented TDM, and nearly all schools (96.6%) report delivering TDM on three or more days per week.” I think the authors mean that of the schools delivering TDM, 96.6% delivered on 3 or more days per week. 2. Background 2nd paragraph – the authors mention a trial and economic analysis conducted in Birmingham – please include a brief sentence on what this study found in terms of effectiveness/cost-effectiveness. 3. Background 3rd paragraph – similarly, please could the authors give some indication of what the qualitative and ethnographic studies show in relation to TDM implementation. 4. Methods – this study is exploring implementation of a complex intervention (TDM). There is no detail about how the survey questions were developed to explore implementation. Did the authors use any implementation theory to inform development of their survey questions? I would suggest that the authors include more detail on how the survey measures were developed and what the survey questions are. 5. Analysis – only analysis of interview data is mentioned, what about the analysis approach to the survey data? 6. Results, Table 1 – left hand column, authors have ‘% in Q1’ but
---

	the reader has to refer to the footnote to know that this relates to deprivation. Can the authors please make this clearer in the body of the table. 7. Table 2 – is there a better way to present this? Perhaps have each year group represented on each table row and then have the % of schools delivering that deliver to that particular age group. This would give slightly different information but would enable the reader to clearly see which age groups are most likely to get TDM and which age groups are least likely to receive it. 8. Survey results -there seems to be very little presentation of these. In the methods it is mentioned that there were 8 questions and these related in part to teachers' experiences of delivery, but the only results presented seem to relate to the school characteristics of implementers and non-implementers, frequency of TDM sessions per week, and which year groups receive it. More detail on survey findings should be presented. 9. Qualitative study sample size – only 5 school representatives were interviewed, despite 19 indicating a willingness to take part. This is a very small number, why were only 5 recruited for this part of the study? The small number of interviewees introduces substantial limitations. 10. Discussion – a fuller discussion of the limitations is needed – only 50% response to survey and then a very small qualitative study, both of which pose substantial limitations. 11. Comparison with other literature – the authors state the adoption rate was 34% here, but I am unsure where this figure has come from. Of 42 schools responding to the survey, 25 had implemented TDM. 12. A key finding seems to be that schools may start off with TDM but they change this physical activity intervention sometimes substantially to suit their particular context. The implications for practice are brought out in the paper, but this finding also has implications for the relevance of any evidence for effectiveness of TDM in a real world setting, as schools are not necessarily delivering this intervention. It would be good to see this discussion point raised.
--	---

REVIEWER	Ram, Bina Imperial College London - Charing Cross Campus, Department of Primary Care and Public Health
REVIEW RETURNED	31-Jan-2021

GENERAL COMMENTS	This paper describes the variability in the implementation of The Daily Mile in the city of Leicester, UK. It is a very interesting paper and highlights some of the difficulties that schools face in carrying out The Daily Mile at their school, which is described as a simple initiative. I would suggest few minor points for clarification. 1. In the introduction or discussion, the authors should include a brief reference to The Daily Mile's 10 core principles (available on TDM Foundation website). There are several important themes identified from this work which can be directly linked back to the principles, i.e., limited space, non-competitive element, weather.
---

	Additionally, under the implications section, the authors discuss the issue of a very important finding - some children's shoes were not suitable for all weather conditions that would impact their participation in The Daily Mile. This could also be referenced back to the core principles. 2. Under analysis it states one researcher carried out the thematic analysis. Ideally there should be two who conduct this independently and come together to discuss. However, in the author contribution statement, it states 'MGH and DH discussed and refined the emerging codes and final themes'. Please add this to the analysis section as it ensures consistency and rigor of the themes identified. 3. Reference #1 – fourth author – please correct to Ram, B (not Bina, R).
--	---

VERSION 1 – AUTHOR RESPONSE

Reviewer 1. Thank you for allowing me the opportunity to review this paper. I read it with interest as the Daily Mile intervention has attracted a lot of interest at the national level. I have a number of points and concerns about the paper, and I have outlined these below. A major concern is the small scale of the study, which creates very substantial limitations.

RESPONSE: Thank you for your time.

COMMENT 1. Abstract – the first sentence does not make sense: “Overall, 37% of schools who completed the survey reported having never implemented TDM, and nearly all schools (96.6%) report delivering TDM on three or more days per week.” I think the authors mean that of the schools delivering TDM, 96.6% delivered on 3 or more days per week.

RESPONSE: Thank you for spotting this. Yes, that is correct and we have adapted that sentence. We also noticed an error in our numbers so we have updated them to match the results within the main paper.

COMMENT 2. Background 2nd paragraph – the authors mention a trial and economic analysis conducted in Birmingham – please include a brief sentence on what this study found in terms of effectiveness/cost-effectiveness.

RESPONSE: This has now been added. We have said “A large (2,280 children from 40 primary schools) randomised controlled trial conducted in Birmingham, England, examined effectiveness of TDM and found no significant difference ($p = 0.146$) in the change in the primary outcome (BMI z-score) between intervention and control schools after 12 months. The accompanying economic analysis did report TDM to be cost-effective.”

COMMENT 3. Background 3rd paragraph – similarly, please could the authors give some indication of what the qualitative and ethnographic studies show in relation to TDM implementation.

RESPONSE: We have added a paragraph to the introduction with more information. We have now added “These studies have identified a range of factors associated with implementation of TDM including simplicity of the intervention, flexibility of delivery, time constraints, weather and the school physical environment, impact on learning time, teacher participation, whole-school delivery, use of goal setting, adaptation and embedding into regular practice.”

COMMENT 4. Methods – this study is exploring implementation of a complex intervention (TDM). There is no detail about how the survey questions were developed to explore implementation. Did the authors use any implementation theory to inform development of their survey questions? I would suggest that the authors include more detail on how the survey measures were developed and what the survey questions are.

RESPONSE: We have now included the survey and interview questions as a supplementary file. The survey was designed pragmatically to gather basic information on the adoption of the programme, the

level of implementation and any barriers to implementation. The intention was to utilise this data to inform the support that could be offered to schools across the city by the Leicester School Sport and Physical Activity Network (SSPAN). As such the survey was not informed by a theoretical framework (e.g. Consolidated Framework for Implementation Research). The interview questions were informed by discussion with the SSPAN and issues noted through informal conversations with local teaching staff, key themes of the RE-AIM framework such as Reach and Maintenance, and the programme components themselves (e.g. delivery of Daily Mile core principles). We have now included a paragraph detailing this within the manuscript.

COMMENT 5. Analysis – only analysis of interview data is mentioned, what about the analysis approach to the survey data?

RESPONSE: We have now added a sentence to confirm that no inferential analysis was done on the data from the survey as they were purely descriptive. On page four we have added “Survey data are presented herein as percentages and ranges. No inferential statistical analyses were conducted on these data.”

COMMENT 6. Results, Table 1 – left hand column, authors have ‘% in Q1’ but the reader has to refer to the footnote to know that this relates to deprivation. Can the authors please make this clearer in the body of the table.

RESPONSE: We have now included IMD in the table and also spelled out IMD fully as index of multiple deprivation in the footnote.

COMMENT 7. Table 2 – is there a better way to present this? Perhaps have each year group represented on each table row and then have the % of schools delivering that deliver to that particular age group. This would give slightly different information but would enable the reader to clearly see which age groups are most likely to get TDM and which age groups are least likely to receive it.

RESPONSE: Thank you for this comment. We see what the reviewer is suggesting. We relooked at the raw data from the original survey and have now added an updated table. We hope this is what the reviewer means.

COMMENT 8. Survey results -there seems to be very little presentation of these. In the methods it is mentioned that there were 8 questions and these related in part to teachers’ experiences of delivery, but the only results presented seem to relate to the school characteristics of implementers and non-implementers, frequency of TDM sessions per week, and which year groups receive it. More detail on survey findings should be presented.

RESPONSE: The free text responses were only there for schools that do not implement TDM fully or at all – it allowed respondents to expand on their answers. We have now added a brief summary of the two main survey free text questions in the results section just before “Stage 2. Qualitative Interview.

COMMENT 9. Qualitative study sample size – only 5 school representatives were interviewed, despite 19 indicating a willingness to take part. This is a very small number, why were only 5 recruited for this part of the study? The small number of interviewees introduces substantial limitations.

RESPONSE: Yes, we were able to secure interview with five of the 19 school leads.

This was disappointing and we did exhaust a number of follow-up ‘chaser’ methods to try and get the rest of the 19 to get involved in the interview. We rang, emailed and even visited the schools within walking distance of the University.

However, we do not feel this is a limitation and stating this as a limitation of our work or saying it is ‘small’ without justification would threaten the validity or generalisability of our study results (see Vasileiou, *et al.* Characterising and justifying sample size sufficiency in interview-based studies: systematic analysis of qualitative health research over a 15-year period. *BMC Med Res Methodol* **18**, 148 (2018) for justification). We feel that code saturation, meaning saturation and alignment with the results of previous qualitative were achieved. The sample size of teachers compares well to the TDM studies we referenced from Ryde *et al* (6 teachers) and Ward *et al* (4 teachers). We have added this into the middle of the limitations section along with supporting references. We were also heartened that we got interviews from two non-implementer schools.

COMMENT 10. Discussion – a fuller discussion of the limitations is needed – only 50% response to survey and then a very small qualitative study, both of which pose substantial limitations.

RESPONSE: As we stated above the number of interviews compares favourably with other TDM studies. Also, saying five interviews is ‘small’ is unjustified when we can argue we have achieved saturation and had rich textured responses. However, in terms of the survey responses yes we agree that 50% is a limitation in terms of quantitative results. We have indeed now stated this in the limitations section. We also say, with references, that availability and access to school staff can be difficult.

COMMENT 11. Comparison with other literature – the authors state the adoption rate was 34% here, but I am unsure where this figure has come from. Of 42 schools responding to the survey, 25 had implemented TDM.

RESPONSE: We have now removed this value because we have decided we cannot extrapolate the numbers we found to the adoption rate for the whole city as our survey response was not for all schools in the city. Instead we have just reported on the adoption rate within our survey but said that this is likely to be much lower. These changes can be seen in the opening line of ‘main findings’ and also in ‘comparison to literature’.

COMMENT 12. A key finding seems to be that schools may start off with TDM but they change this physical activity intervention sometimes substantially to suit their particular context. The implications for practice are brought out in the paper, but this finding also has implications for the relevance of any evidence for effectiveness of TDM in a real world setting, as schools are not necessarily delivering this intervention. It would be good to see this discussion point raised.

RESPONSE: Thank you, this is a great point. We have now added this to the opening of the ‘implications for practice’ section. We have now added “It is clear that adaptations and alternatives to TDM are happening. This has implications for TDM evidence base as well as practice. Schools may begin using TDM but adapt or change their approach, often quite substantially, to suit their particular context. This has implications for the relevance of any evidence for effectiveness of TDM in a real-world setting, as some schools may not necessarily be delivering TDM even if using TDM nomenclature.”

Reviewer 2. This paper describes the variability in the implementation of The Daily Mile in the city of Leicester, UK. It is a very interesting paper and highlights some of the difficulties that schools face in carrying out The Daily Mile at their school, which is described as a simple initiative.

RESPONSE: Thank you for your time.

COMMENT 1. In the introduction or discussion, the authors should include a brief reference to The Daily Mile’s 10 core principles (available on TDM Foundation website). There are several important themes identified from this work which can be directly linked back to the principles, i.e., limited space, non-competitive element, weather. Additionally, under the implications section, the authors discuss the issue of a very important finding - some children’s shoes were not suitable for all weather conditions that would impact their participation in The Daily Mile. This could also be referenced back to the core principles.

RESPONSE: Thank you for this comment. We had mentioned this within the results section but we have also made a clearer reference to it in the discussion page 12. We have added “These themes shed light on how TDM core principles of TDM being quick and kept simple may not always be possible.”

In relation to the shoe point we have now included this in the implication section and highlighted the difficulty that a child in poor quality shoes may have in a muddy field. We have added “While a TDM core principle is that children should not need specialist kit and can do it in their school clothes it is conceivable that children in light or low-quality shoes may find their mile on wet tarmac uncomfortable. Equally, TDM risk assessments state that children in unsuitable footwear should walk (23) which also has implications for the pupil’s own experience and when evaluating the TMD at the pupil level as there are different health and social effects to running and walking a mile.”

COMMENT 2. Under analysis it states one researcher carried out the thematic analysis. Ideally there should be two who conduct this independently and come together to discuss. However, in the author contribution statement, it states ‘MGH and DH discussed and refined the

emerging codes and final themes'. Please add this to the analysis section as it ensures consistency and rigor of the themes identified.

RESPONSE: We have updated this section now to better describe what was done. MGA performed the analysis, DH review the interview transcripts and then themes were discussed and redefined in collaboration.

COMMENT 3. Reference #1 – fourth author – please correct to Ram, B (not Bina, R).

RESPONSE: We have corrected this.

In addition, we have added some new references that have been published since our first submission. We have also updated the wording of the funding statement.

VERSION 2 – REVIEW

REVIEWER	Ram, Bina Imperial College London - Charing Cross Campus, Department of Primary Care and Public Health
REVIEW RETURNED	06-Apr-2021
GENERAL COMMENTS	The authors have clearly addressed all the points raised by the reviewers. I am happy to recommend this paper for publication.